# The Clinical Experience of Mycobacterial Culture Yield of Pleural Tissue by Pleuroscopic Pleural Biopsy among Tuberculous Pleurisy Patients

**DOI:** 10.3390/medicina58091280

**Published:** 2022-09-15

**Authors:** Chung-Shu Lee, Li-Chung Chiu, Chih-Hao Chang, Fu-Tsai Chung, Shih-Hong Li, Chun-Liang Chou, Chih-Wei Wang, Shu-Min Lin

**Affiliations:** 1Department of Thoracic Medicine, School of Medicine, Chang Gung Memorial Hospital, Chang Gung University, Taipei 105406, Taiwan; 2Department of Pulmonary and Critical Care Medicine, New Taipei Municipal Tucheng Hospital, New Taipei City 236017, Taiwan; 3Department of Respiratory Therapy, New Taipei Municipal Tucheng Hospital, New Taipei City 236017, Taiwan; 4Department of Thoracic Medicine, School of Medicine, Taipei Medical University Hospital, Taipei Medical University, Taipei 110301, Taiwan; 5Department of Anatomic Pathology, Chang Gung Memorial Hospital, Linkou Branch, Taoyuan 333423, Taiwan; 6Department of Respiratory Therapy, College of Medicine, Chang Gung Memorial Hospital, Chang Gung University, Taoyuan 333423, Taiwan

**Keywords:** tuberculosis, pleuroscope, mycobacterial culture

## Abstract

*Background and Objectives:* Tuberculous pleurisy is a common extrapulmonary TB that poses a health threat. However, diagnosis of TB pleurisy is challenging because of the low positivity rate of pleural effusion mycobacterial culture and difficulty in retrieval of optimal pleural tissue. This study aimed to investigate the efficacy of mycobacterial culture from pleural tissue, obtained by forceps biopsy through medical pleuroscopy, in the diagnosis of TB pleurisy. *Materials and Methods:* This study retrospectively enrolled 68 TB pleurisy patients. Among them, 46 patients received semi-rigid pleuroscopy from April 2016 to March 2021 in a tertiary hospital. We analyzed the mycobacterial culture from pleural tissue obtained by forceps biopsy. *Results:* The average age of the study participants was 62.8 years, and 64.7% of them were men. In the pleuroscopic group, the sensitivity of positive *Mycobacterium tuberculosis* (M. TB) cultures for sputum, pleural effusion, and pleural tissue were 35.7% (15/42), 34.8% (16/46), and 78.3% (18/23), respectively. High sensitivities of M. TB culture from pleural tissue were up to 94.4% and 91.7% when pleural characteristic patterns showed adhesion lesions and both adhesion lesions and presence of micronodules, respectively. *Conclusions:* M. TB culture from pleural tissue should be considered a routine test when facing unknown pleural effusion during pleuroscopic examination.

## 1. Introduction

Tuberculosis (TB) remains a serious infectious cause of death worldwide, with 5.8 million new cases and 1.3 million deaths in 2020 [1], despite aggressive public health strategies and effective treatment. The Bureau of National Health Insurance (NHI) in Taiwan introduced the no-notification-no-reimbursement policy in 1997 to enhance TB notification and to decrease the incidence of TB. Accordingly, the NHI, which has enrolled up to 99% of Taiwanese, has, since 2007, reached a TB notification rate of more than 97%. The directly observed treatment short-course (DOTS) program implementation and active contact investigation started in 2006 [2]. Taiwan is a TB-endemic area with an intermediate TB burden. The TB incidence in Taiwan dropped from 72.5 cases per 100,000 population in 2005 to 43.9 cases per 100,000 population in 2016 [3]. Pulmonary TB is the most common form of TB, with extrapulmonary TB accounting for up to 15% of the prevalence [4]. The incidence of TB pleurisy as a manifestation of extrapulmonary TB is approximately 23–66% [5]. Moreover, pleural TB is the most common cause of exudative effusion, accounting for 25–44% of all cases of pleural effusion [6].

Diagnosis of TB pleurisy depends on microscopic examination of the pleural fluid for acid-fast bacilli (AFB), mycobacterial culture of the sputum, pleural fluid, or pleural tissue, and pathological examination of the pleural tissue for epithelioid cell granuloma and/or caseous granuloma [7]. Microscopic examination of the pleural fluid is positive for AFB in less than 5% of patients because of the specific paucibacillary nature of the disease [8,9], whereas mycobacterial culture of the pleural fluid has low sensitivity (24–58%) [9,10]. At present, both biopsy of the pleural tissue for histological results and mycobacterial culture of the pleural fluid or tissue are the most sensitive among the available methods [11]. Recent evidence suggests that whole genome sequencing (WGS) of M. TB can be used as an alternative molecular method for TB diagnosis [12] and detection of drug resistance [13]. Application of WGS was also extended to extrapulmonary sites, including TB pleurisy [14].

In clinical practice, closed pleural biopsy or surgical biopsy are commonly used to obtain pleural tissue for pathological examination. However, there is low sensitivity for diagnosing TB pleurisy, ranging from 50% to 60% [11]. Recent evidence has revealed that semi-rigid pleuroscopy-guided biopsy is an effective diagnostic tool for undiagnosed pleural effusion [15]. Semi-rigid pleuroscopy was conducted by a non-surgical pulmonologist in the endoscopy suite, with patients under local anesthesia and conscious sedation. This procedure offers the benefit of a high diagnostic yield via direct visualization of pleural lesions for biopsy and low complication rates [15]. In addition to obtaining tissue for pathological studies, pleuroscopic biopsy can also obtain tissue for mycobacterial culture. The results of mycobacterial cultures from pleural tissue can provide a sensitivity test for anti-tuberculosis drugs. Drug sensitivity information is crucial to the success rate of TB treatment, especially in countries with high drug resistance rates [16]. However, few studies have reported the results of mycobacterial cultures from tissue obtained by pleuroscopic biopsy for pleural TB.

Apart from its role in improving the diagnostic yield of undiagnosed pleural effusion, abnormal pleural lesions can be observed directly under a microscope. Several characteristic patterns were observed during the pleuroscopic examination [7]. By identifying the characteristic features of TB pleurisy, precise biopsy of the targeted lesion may increase the diagnostic yield rate. In addition, tissue culturing for mycobacterial culture for mycobacterial detection and drug sensitivity tests can be performed using this procedure. The relationship between these characteristic features and the yield of pleural mycobacterial cultures has never been investigated. 

The aim of the present study was to investigate the diagnostic yield of mycobacterial cultures from pleural tissue obtained by pleuroscopic forceps biopsy for TB pleurisy. In addition, the study also attempted to determine the relationship between pleuroscopic features and the incidence of positive mycobacterial cultures of pleural tissue.

## 2. Materials and Methods

### 2.1. Patients

The study was approved by the Institutional Review Board of the Chang Gung Memorial Foundation (IRB No. 202100712B0). We retrospectively recruited 247 patients from April 2016 to March 2021 at Linkou Chang Gung Memorial Hospital, Taiwan, which is a tertiary referral medical center. Consecutive patients who underwent semi-rigid pleuroscopy within the study period were identified from medical records, and their demographic characteristics were collected. Patients received pleuroscopic study due to undiagnosed pleural effusion after initial pleural effusion aspiration study. Among the 247 patients receiving pleuroscopy, 109 patients were diagnosed with malignancy, 60 patients had fibrinous pleuritis, 17 patients had other diagnoses, 11 patients were undiagnosed, and 4 patients were excluded due to failure in biopsy. Subsequently, a total of 46 TB pleurisy patients were recruited for the final analysis (Figure 1).

### 2.2. Diagnostic Criteria of TB Pleurisy

Patients with TB pleurisy were defined according to a previous study. The study included patients with confirmed and probable TB pleurisy [17]. Patients with confirmed TB pleurisy were defined as patients with pleural fluid samples that were culture-positive for *Mycobacterium tuberculosis* (M. TB) and/or a histopathological finding consistent with TB by pleural biopsy. Patients with probable TB pleurisy were defined as those with at least one of the following circumstances: (1) sputum specimens that were culture-positive for *M. tuberculosis*, (2) other biological specimens that were culture-positive for *M. tuberculosis*, (3) elevated pleural lymphocyte count and protein levels, or (4) a positive response to anti-tuberculosis medication with absence of other possible causes of pleural effusion. The present study adopted a combination of pleural *M. tuberculosis* culture, pleural pathologic analysis, and clinical diagnosis as the reference methods for diagnosing TB pleurisy, which was independently conducted by two senior physicians.

The definition of primary and reactivation TB was defined according to chest radiographic criteria [18]. Tuberculous pleurisy associated with reactivation TB was defined as patients whose chest radiographs revealed infiltrates with or without cavitation in the upper lobes or the apical segment of either lower lobe, or the presence of parenchymal scarring in the upper lobes suggestive of previous TB. Pleural TB was considered to be a sequel of primary disease in patients, whose chest radiographs did not show any parenchymal infiltrate.

### 2.3. Procedure and Equipment

In all patients, a pleuroscopic examination was conducted by an experienced operator and two trained assistants. Additionally, during the procedure, all patients received moderate sedation with fentanyl and midazolam, with or without propofol. Before the initiation of the procedure, thoracic ultrasonography was performed to determine the optimal entry point. After local administration of 2% lidocaine for anesthesia, a 1.0 cm skin incision was made followed by thoracotomy with an 8-mm flexible trocar (MAJ-1058, Olympus Medical Systems Corp., Tokyo, Japan). A semi-rigid thoracoscope (LTF-240; Olympus, Tokyo, Japan) was used to drain the effusion initially and explore the pleural cavity. The accessible pleural space was first entirely visualized, and the biopsy site determined. Forceps (FB-15C-1; Olympus) were used for biopsy through the working channel of the thoracoscope. At least three biopsies were routinely performed for each patient. The biopsy area was restricted to the parietal pleura. For histopathological analyses, all biopsy samples were transported in separate formalin containers. Biopsy samples were processed according to the standard protocols for histopathology and immunohistochemical staining. After the biopsies were completed, a drainage tube with a 16-Fr. pigtail catheter (BT-PDS-1630-W-NK1; Bioteq, Taipei, Taiwan) was placed into the pleural cavity to monitor and drain the pleural effusion. The patients received a follow-up chest X-ray radiograph 3 h after the procedure to ensure the location of the drainage tube. The next follow-up chest X-ray radiograph was performed 3 days later. If follow-up chest radiography did not reveal pneumothorax, and the daily drainage amount was less than 50 mL, the catheter was removed.

### 2.4. Specimen Processing

The processing of pleural specimens was managed using the procedure described in a previous report in [19]. Briefly, pleural fluid, tissue specimens, and sputum specimens were processed with Ziehl-Neelsen stain for microscopic examination. Decontamination was performed using N-acetyl-L-cysteine-sodium hydroxide (BBL MycoPrep; Becton Dickinson, Cockeysville, MD, USA).

### 2.5. Drug Susceptibility Testing (DST)

The mycobacterial laboratory of our institute is one of the Taiwan Center of Disease Control (CDC)-contracted clinical TB laboratories in Taiwan. The participating laboratories used the routine DST methods under the regulation and surveillance of Taiwan CDC. The agar proportion method on Middle-brook 7H10 (Creative Microbiologicals or Sancordon, Taipei, Taiwan) and BACTEC MGITTM 960 SIRE Kits with a liquid culture system were recommended. The critical drug concentrations for the agar proportion method on 7H10 were INH 0.2 μg/mL, INH 1.0 μg/mL, RMP 1.0 μg/mL, EMB 5.0 μg/mL and SM 2.0 μg/mL, while on MGIT 960 system, the recommended critical concentrations were INH 0.1 μg/mL, RMP 1.0 μg/mL, EMB 5.0 μg/mL and SM 1.0 μg/mL. Growth on the control medium was compared with the growth on the drug-containing medium to determine susceptibility. DST results were classified as resistant or susceptible. Tests were validated by the susceptibility of *M. tuberculosis* H37Rv included in the same DST.

### 2.6. Statistical Analysis

Data are expressed as mean ± standard error of the mean. Categorical variables were compared using the chi-squared test or Fisher’s exact test. The odds ratios (ORs) and 95% confidence intervals (CIs) were compared. Statistical significance was set at *p* < 0.05. Statistical analyses were performed using SPSS (version 13.0; SPSS Inc., Chicago, IL, USA).

## 3. Results

This study included 46 patients with pleural TB who were admitted to the Linkuo branch of the Chang Gung Memorial Hospital in Taiwan between April 2016 and March 2021. None of the patients had disseminated TB. Among the 46 pleural TB patients, 5 (10.9%) patients were secondary TB and the other 41 (89.1%) patients were primary TB. Under medical semi-rigid pleuroscopy, the patients underwent pleural forceps biopsy for pleural tissue. In the study, all the 46 pleural TB patients received pleuroscopic pleural biopsy. M. TB culture study was performed on the sputum of 42 patients, the pleural effusion of 46 patients, and the pleural tissue of 23 patients (Figure 1). Histopathologic findings consistent with TB by pleural biopsy were found in 21 of 27 patients with negative sputum culture, 20 of 30 patients with negative pleural effusion culture, and 5 patients with negative pleural tissue culture. Demographic data of the patients are listed in Table 1. The average age of all patients was 62.5 years, among which the percentage of men was 65.2%. All patients qualified for exudative pleural effusion. High lymphocyte count percentage and LDH levels in the pleural effusions were noted at 87.5% and 309 mg/dL, respectively. 

In this study, sensitivity was calculated using clinical diagnosis as the reference: sputum culture yielded positive M. TB results in 35.7% (15/42), pleural effusion culture yielded positive M. TB results in 34.8% (16/46), and 78.3% of pleural TB patients (18/23) were positive for M. TB in the pleural tissue (Figure 1). The sensitivity of histological examinations of tissue samples obtained by medical thoracoscopy for the detection of pleural TB was 95.6% (44/46). Two patients were diagnosed with pleural TB only by positive M. TB culture in the pleural tissue. The diagnostic yields of cultures from pleural tissue were significantly higher than those from sputum (OR, 6.48; 95% CI 2.07–18.65; *p* = 0.001) and pleural effusion (OR, 6.75; 95% CI 2.01–19.04; *p* = 0.0007) M. TB cultures, respectively (Figure 1). In 18 patients with positive M. TB culture results from pleural tissue, only 6 (33.3%) patients were sputum positive for M. TB cultures and 7 patients (38.9%) were pleural effusion positive for M. TB cultures.

### 3.1. TB Pleurisy Diagnosis by M. TB Cultures and Pathology Results

Among the 46 cases of TB pleurisy, positive M. TB cultures were obtained from sputum (*n* = 15), pleural effusion (*n* = 16), and pleural tissues (*n* = 18) (Table 2). Positive AFB smear from pleural tissue was found in only one patient with sputum and effusion positive for M. TB cultures. However, no patient with AFB was found among those with pleural tissue positive for M. TB cultures. The pathological results revealed epithelioid cell granulomas with or without caseous granuloma (Table 2). Two patients were diagnosed with pleural TB by positive pleural tissue for M. TB cultures without the presence of pathological features of tuberculosis.

### 3.2. TB Pleurisy Diagnosis by M. TB Cultures and Pleuroscopic Features

Besides the culture assays, the study further compared the sensitivity of M. TB cultures based on these pleuroscopic features (Figure 2). The sensitivity of M. TB cultures from sputum, pleural effusion, and pleural tissues was 31.3%, 29.4%, and 70.6%, respectively, in TB pleurisy patients with micronodules under pleuroscopic examination (Table 3). The sensitivity of M. TB cultures from sputum, pleural effusion, and pleural tissues was 48.0%, 42.4%, and 94.4%, respectively, in TB pleurisy patients with adhesion lesions under pleuroscopic examination. The sensitivity of TB cultures from sputum, pleural effusion, and pleural tissues was 27.3%, 36.4%, and 91.7%, respectively, in TB pleurisy patients with both micronodules and adhesion lesions under pleuroscopic examination.

### 3.3. Drug Sensitivity of Mycobacterial Cultures from Pleural Tissue

In the 18 patients with positive pleural tissue for M. TB cultures, the drug susceptibility revealed that all cultures were sensitive to ethambutol, ofloxacin, rifampin, and higher dose isoniazid (1.0 μg/mL). However, drug resistance to streptomycin and lower dose isoniazid (0.2 μg/mL) was found in one (5.6%) and two (11.1%) patients, respectively.

### 3.4. Complications and Outcomes

Neither moderate nor severe bleeding was observed during or following the clinical course. One patient (2.2%) developed mild bleeding. No subcutaneous emphysema, pneumomediastinum, pneumothorax, or wound infection developed in either group. None of the patients with TB required thoracic surgery.

## 4. Discussion

To provide pathological evidence for the diagnosis of TB pleurisy, a pleuroscopic biopsy could obtain adequate tissue for M. TB culturing. The study demonstrated that the positive yield of pleural tissue M. TB culture was higher than that of M. TB culture of sputum and pleural fluid. Pleuroscopic findings of micronodules or adhesions were associated with high positive rates of pleural tissue for M. TB cultures. Apart from providing M. TB culture to confirm the diagnosis of pleural TB, the drug sensitivity test also offered important information for the adjustment of anti-TB drug selection.

Among all included patients, the positive sputum and pleural effusion TB culture rates were 35.7% and 34.8%, respectively. Meanwhile, the study demonstrated a high positive culture rate for pleural tissue of up to 78.3%. Among 23 TB pleurisy patients, two reached the diagnosis with positive pleural M. TB culture as the only clue for diagnosis, without any granulomatous inflammation on pathological examination. Therefore, mycobacterial culturing from pleuroscopic biopsy tissue may improve the diagnostic yield of TB pleurisy. Evidence suggests that several reasons account for the failure of TB therapy, such as late diagnosis and lack of timely and proper administration of effective drugs [20,21]. The correct diagnosis of TB pleurisy is critical to clinical outcomes. TB is a potentially curable disease if accurate diagnosis can be achieved and effective anti-tuberculosis treatment is administered, as prognosis is worse if anti-tuberculosis treatment is delayed after the initial visit [22]. Unfortunately, conventional diagnostic tests are limited in clinical use. Microscopy of the pleural fluid for AFB is positive in less than 5% of patients, due to the pauci-bacillary nature of the disease [8,9], whereas mycobacterial culture of the pleural fluid is time-consuming and has low sensitivity (24–58%) [9,10]. The levels of adenosine deaminase (ADA) in pleural fluid have high sensitivity and specificity in the diagnosis of TB pleurisy [23,24]. However, the use of ADA may not be widely available in developing countries and cannot provide information on drug sensitivity. Pleuroscopic biopsy of pleural tissue for combined histological examination and mycobacterial culture of the pleural fluid and tissue is the most sensitive of the currently available diagnostic methods.

In another similar study, [25], the microbiological yield of the pleural tissue culture on thoracoscopic biopsy sample was 39%. However, our result showed a yield of tissue culture up to 78.3%. The possible cause of the higher yield in our study may be related to selection of biopsy site according to the specific features favoring TB pleurisy [7]. In the previous study, biopsy was performed from the visualized abnormal areas, and in patients with no visualized pleural abnormality, it was taken from randomized areas in the parietal pleura.

The Xpert MTB/RIF assay (Xpert; Cepheid, Sunnyvale, CA, USA) is a rapid, WHO endorsed, automated PCR test optimized for respiratory specimens that can detect both M. TB and rifampicin resistance. Previous studies have demonstrated that Xpert MTB/RIF assay in pleural tissue provided a higher yield than culture [25,26]. Further study is needed to investigate the role of combination of pleuroscopic TB pleurisy features and Xpert MTB/RIF in the diagnostic yield of TB pleurisy.

Compared with closed pleural biopsy [27], pleuroscopic forceps biopsy has some advantages. Under a pleuroscope, operators can visualize the pleural lesion with precise biopsy instead of a blind closed pleural biopsy to obtain pathological specimens [15]. The improved diagnostic yield of pleural tissue in pathological examination and M. TB culture by semi-rigid pleuroscopy is mostly due to examination of the pleural lesion and guiding of the biopsy for abnormal pleural lesions. Several characteristic patterns were observed during the pleuroscopic examination. The study showed that the presence of specific features, such as micronodules and adhesions, was associated with high positive culture rates. By identifying the characteristic features of TB pleurisy, precise biopsy of the targeted lesion may increase the diagnostic yield rate and reduce the duration of the procedure [7,28]. Pleuroscopic biopsy is commonly used in undiagnosed pleural effusion after an initial pleural effusion aspiration study. Among the undiagnosed pleural effusions, malignant cancers, rather than TB, are the most popular [15]. Therefore, pleural tissue culture was not routinely performed in all the recruited patients of this study. According to our results, routine pleural tissue culture for M. TB in undiagnosed pleural effusion patients receiving pleuroscopic biopsy may be necessary. Considering the cost effectiveness of M. TB culture, pleural tissue M. TB culture may be mandatory in countries with moderate or high TB prevalence. In countries with low TB prevalence, pleural tissue M. TB culture may be performed in patients with these specific features favoring TB pleurisy during a pleuroscopic study.

Recent global surveys have reported that drug-resistant TB exists in every location [29]. The World Health Organization reported a trend toward an increasing number of cases of drug-resistant TB, which has further aggravated the challenges of an already lengthy and complicated treatment course [30]. Our study showed that isoniazid (INH) resistance was observed in more than 10% of cultures from pleural tissue. INH is an important first-line agent for the treatment of TB because of its potent early bactericidal activity. However, resistance to INH is very common, with a prevalence rate of 28% among previously treated cases, and 10% among new cases [31]. In this study, cultures from tissue for mycobacterial detection and drug sensitivity tests were performed after pleuroscopic biopsy. With drug sensitivity results from the cultures, aggressive treatment may be administered to improve the outcome while avoiding drug resistance.

There were some limitations to this study. First, this was a retrospective study. Second, the study specimen numbers were small, even after five-year data collection. Finally, acquiring adequate specimens for tissue culture depends on the technique used by skillful pulmonologists. However, this is the first report evaluating the diagnostic yield and drug sensitivity of M. TB culture from pleural tissue obtained by pleuroscopy. Lastly, the study recruited patients with undiagnosed pleural effusion after initial pleural effusion aspiration study. In patients with pleural effusion, aspiration of pleural effusion should be the first step of study. Pleuroscopy was used only in certain subpopulations or in the hospital without pleural fluid ADA measurement. Therefore, the results of this study should be interpreted with caution. Our study provides a basis for future prospective studies to evaluate the role of pleuroscopic biopsy in the diagnosis and treatment outcomes of patients with TB pleurisy.

## 5. Conclusions

This study revealed a high diagnostic yield (78.3%) of M. TB culture from pleural tissue obtained by pleuroscopic forceps biopsy. Routine mycobacterial culturing from sputum, pleural effusion, and pleural tissue may be recommended in patients receiving pleuroscopic study due to their undiagnosed pleural effusion. In addition, if any characteristic features can be observed on pleuroscopy, the operator should obtain specimens from multiple sites for mycobacterial culturing, as well as for pathological examination.

## Figures and Tables

**Figure 1 medicina-58-01280-f001:**
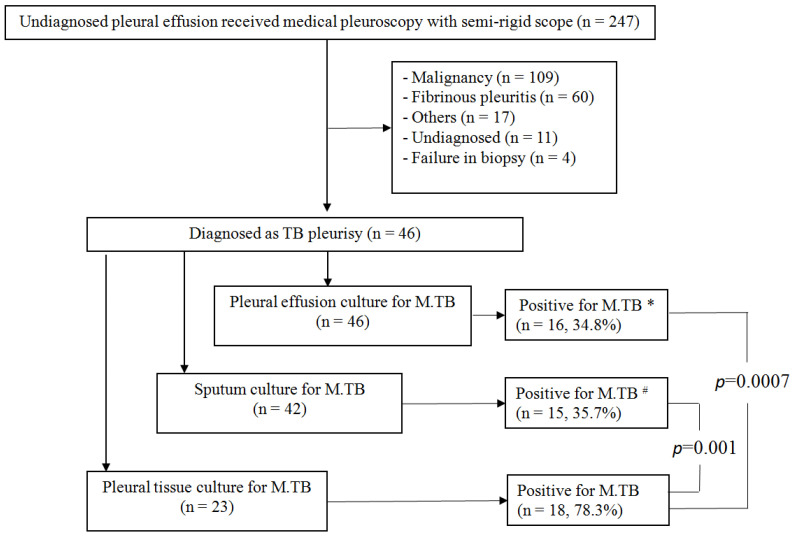
The diagnosis of patients that received pleuroscopic forceps biopsy; *, #: *p* < 0.05, statistical significance.

**Figure 2 medicina-58-01280-f002:**
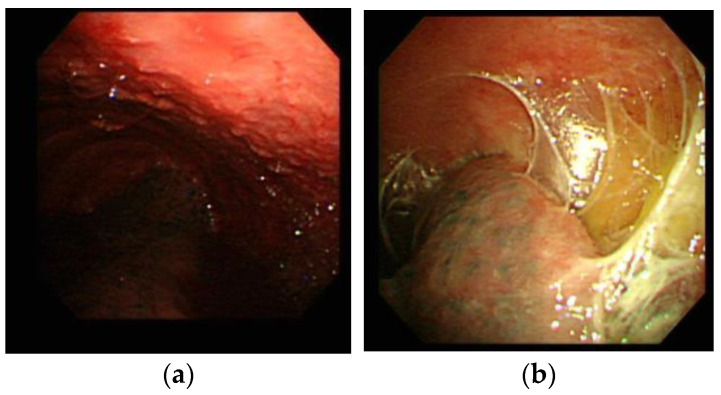
(**a**) A representative case of tuberculous pleurisy. The pleuroscopic image shows micronodules; (**b**) A representative case of an adhesion lesion. The pleuroscopic image shows the presence of fibrous bands between the visceral and parietal pleura.

**Table 1 medicina-58-01280-t001:** Baseline characteristics of the TB pleurisy patients.

Variables	Total *n* = 46
Age, median (IQR)	62.5 (20.4–92.0)
Male Gender	30 (65.2%)
Smoking status	
Current smoker	3 (6.5%)
Prior smoker	11 (23.9%)
Never smoker	32 (69.6%)
Location of pleural effusion	
Right	28 (60.9%)
Left	13 (28.3%)
Bilateral	5 (10.9%)
Pleural effusion lymphocyte percentage (%), median (IQR) *	87.5% (76.0–95.0%)
Pleural effusion LDH (mg/dL), median (IQR)	309 (229.5–405.5)
Pleural effusion total protein (mg/dL), median (IQR)	5.05 (4.2–5.4)
Serum LDH (mg/dL), median (IQR)	199 (178–249)
Serum total protein (mg/dL), median (IQR)	6.95 (6.5–7.7)
Pleural effusion/Serum LDH ratio	1.40 ± 0.54
Pleural effusion/Serum total protein ratio	0.68 ± 0.11

* IQR: interquartile range.

**Table 2 medicina-58-01280-t002:** Positive TB cultures of the TB pleurisy patients by pathology classification.

Pathology Results	Positive Mycobacterial Culture in		
	Sputum *n* = 15	Pleural Fluid *n* = 16	Pleural Tissue Biopsy * *n* = 18
Acid-fast bacillus in specimens	1 (6.7%)	1 (6.3%)	0 (0.0%)
Epithelioid cell granuloma	8 (53.3%)	6 (37.5%)	6 (33.3%)
Caseous granuloma + epithelioid cell granuloma	7 (46.7%)	10 (62.5%)	10 (55.6%)

* Two patients were diagnosed as pleural TB by pleural tissue culture without the presence of pathological features of tuberculosis.

**Table 3 medicina-58-01280-t003:** TB cultures of the TB pleurisy patients by pleuroscopic classification.

Visual Characteristic by Pleuroscopy	Positive Mycobacterial Cultures/Features under Pleuroscope		
	Sputum	Pleural Fluid	Pleural Tissue Biopsy
With micronodules	10/32 (31.3%)	10/34 (29.4%)	12/17 (70.6%)
With adhesion lesions	11/31 (48.0%)	14/33 (42.4%)	17/18 (94.4%)
With combination of micronodules and adhesion lesions	6/22 (27.3%)	8/22 (36.4%)	11/12 (91.7%)

## Data Availability

The data sources in this study are available from the corresponding author on rational request.

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
