# Peer review of "The Clinical Experience of Mycobacterial Culture Yield of Pleural Tissue by Pleuroscopic Pleural Biopsy among Tuberculous Pleurisy Patients"

_medicina, 2022, doi:10.3390/medicina58091280_

Round 1
Reviewer 1 Report
Authors have described their retrospective analysis of undiagnosed pleural effusions which turned out to be due to tuberculosis. My few comments.
1. The section on 'Discussion' should be improved. Authors should compare their results with other similar studies. Is the yield in their study different from other studies, if yes then what is the reason? For example they could compare their results with the study https://pubmed.ncbi.nlm.nih.gov/29486527/.
2. What is the role of Xpert MTB/RIF assay in such patients? Was it done in this study?
Author Response
Reviewer 1
Authors have described their retrospective analysis of undiagnosed pleural effusions which turned out to be due to tuberculosis. My few comments.
(1) The section on 'Discussion' should be improved. Authors should compare their results with other similar studies. Is the yield in their study different from other studies, if yes then what is the reason? For example they could compare their results with the study https://pubmed.ncbi.nlm.nih.gov/29486527/.
Reply: Thanks for your valuable comment. We had added a description as “In other similar study [28], the microbiological yield of the pleural tissue culture on thoracoscopic biopsy sample was 39%. However, our result showed a yield of tissue culture up to 78.3%. The possible cause of the higher yield in our study may be related to selection of biopsy site according to the specific features favoring TB pleurisy [7]. In the previous study, biopsy was performed from the visualized abnormal areas, and in patients with no visualized pleural abnormality, it was taken from randomized areas in the parietal pleura.” In page 7, line 272 to 278.
(2)What is the role of Xpert MTB/RIF assay in such patients? Was it done in this study?
Reply: We did not perform Xpert MTB/RIF assay because the device is not available in our institute. But we agree that using Xpert MTB/RIF assay in pleural tissue may provide important information. Therefore, we have added sentences as “The Xpert MTB/RIF assay (Xpert; Cepheid, Sunnyvale, CA, USA) is a rapid, WHO endorsed, automated PCR test optimized for respiratory specimens that can detect both M. TB and rifampicin resistance. Previous studies have demonstrated that Xpert MTB/RIF assay in pleural tissue provided a higher yield than culture [28, 29]. Further study is needed to investigate the role of combination of pleuroscopic TB pleurisy features and Xpert MTB/RIF in the diagnostic yield of TB pleurisy.” in page 7, line 279 to 284.

Reviewer 2 Report
Lee et. al. have aimed theirs research on diagnostics of extrepulmonary TB from biopsy material. The results showed high sensitivity in the diagnosis of tuberculous pleuritis from from pleural tissue when pleural characteristic patterns showed adhesion lesions and both adhesion lesions and presence of micronodules. These results have significant clinical potential, however, I have some comments that will need to be clarified. Based on the comments below, I encourage authors to make a few edits before publishing:
Comments:
Line 38-39 – there is an information about TB incidence and deaths in 2019. Please, update these information based on the newest Global TB report published by WHO.
Line 59 – I strongly encourage authors to mention also the molecular methods, primarily WGS, which can be used in diagnostics of pulmonary and extrapulmonary-TB (including TB pleurisy) and has a wide range of uses, including identification of complete resistance profile (citations: 10.2147/IDR.S269779; 10.1038/s41598-022-11287-5; 10.1089/omi.2017.0070). Moreover, WGS can be performed directly from clinical specimens.
Have you considered using the conventional XPert MTB/RIF molecular method for diagnosis? This method is more sensitive than histopathology and other culture methods.
Line 164 – why you didnt performed the pleural tissue cultivation for all patients?
Line 166 – It is not necessary to mention the beginning of the sentence „In patients with negative M. TB culture results“ , as you further mention that it is about culture negative patients.
Line 216-222 – Table 4. it is unnecessary to attach, you explained everything clearly in the text. You didnt mention the CC of fluoroquinolones. Also the most important is to mention the medium used for pDST. There are different CC of first- and second-line antiTB drugs for MGIT, LJ, 7H10 etc, therefore, it is essential to mention this information to confirm resistance. How did you choose the drugs tested? There is not any information about pyrazinamid.
Line 216-222 – It is very uncommon that Mtb strain is resistant to lower concentration of INH and sensitive to higher concentration. Do you have any explanation?
Line 230 – The sentence does not have to start with the word "In addition"
Author Response
Reviewer 2
Comments:
- Line 38-39 – there is an information about TB incidence and deaths in 2019. Please, update these information based on the newest Global TB report published by WHO.
Reply: In the revised introduction section, we have updated the information as “Tuberculosis (TB) remains a serious infectious cause of death worldwide, with 5.8 million new cases and 1.3 million deaths in 2020” in line 38 to 39.
2.Line 59, Part I – I strongly encourage authors to mention also the molecular methods, primarily WGS, which can be used in diagnostics of pulmonary and extrapulmonary-TB (including TB pleurisy) and has a wide range of uses, including identification of complete resistance profile (citations: 10.2147/IDR.S269779; 10.1038/s41598-022-11287-5; 10.1089/omi.2017.0070). Moreover, WGS can be performed directly from clinical specimens.
Reply: We had added sentences from Line 61 to 64 as “Recent evidence suggests that whole genome sequencing (WGS) of M. TB can be used as an alternative molecular method for TB diagnosis [12] and detection of drug resistance [13]. Application of WGS was also extended to extrapulmonary sites including TB pleurisy [14].”
Part II: Have you considered using the conventional XPert MTB/RIF molecular method for diagnosis? This method is more sensitive than histopathology and other culture methods.
Reply: We did not perform Xpert MTB/RIF assay because the device is not available in our institute. But we agree that using Xpert MTB/RIF assay in pleural tissue may provide important information. Therefore, we have added sentences as “The Xpert MTB/RIF assay (Xpert; Cepheid, Sunnyvale, CA, USA) is a rapid, WHO endorsed, automated PCR test optimized for respiratory specimens that can detect both M. TB and rifampicin resistance. Previous studies have demonstrated that Xpert MTB/RIF assay in pleural tissue provided a higher yield than culture [28, 29]. Further study is needed to investigate the role of combination of pleuroscopic TB pleurisy features and Xpert MTB/RIF in the diagnostic yield of TB pleurisy.” in page 8, line 279 to 284.
3.Line 164 – why you didn’t performed the pleural tissue cultivation for all patients?
Reply: Pleuroscopic biopsy is commonly used in undiagnosed pleural effusion after an initial pleural effusion aspiration study. Among the undiagnosed pleural effusions, malignant cancers, rather than TB, are the most popular [15]. Therefore, pleural tissue culture was not routinely performed in all the recruited patients of this study. According to our results, routine pleural tissue culture for M. TB in undiagnosed pleural effusion patients receiving pleuroscopic biopsy may be necessary. To clarify this point, we have added the above description in the revised discussion section, page 8, line 295 to 300.
4.Line 166 – It is not necessary to mention the beginning of the sentence „In patients with negative M. TB culture results“, as you further mention that it is about culture negative patients.
Reply: Thanks for your comment. We had deleted “In patients with negative M. TB culture results’ in the revised manuscript.
5.Line 216-222 – Table 4. it is unnecessary to attach, you explained everything clearly in the text. You didnt mention the CC of fluoroquinolones. Also the most important is to mention the medium used for pDST. There are different CC of first- and second-line antiTB drugs for MGIT, LJ, 7H10 etc, therefore, it is essential to mention this information to confirm resistance. How did you choose the drugs tested? There is not any information about pyrazinamid.
Reply: We have deleted table 4 in the revised manuscript. The mycobacterial laboratory of our institute is one of the Taiwan CDC-contracted clinical TB laboratories in Taiwan. The participating laboratories used the routine DST methods under the regulation and surveillance of Taiwan CDC. In line 156 to 168 of the revised materials and methods section, we have added 2.5 Drug susceptibility testing (DST)
The mycobacterial laboratory of our institute is one of the Taiwan Center of Disease Control (CDC)-contracted clinical TB laboratories in Taiwan. The participating laboratories used the routine DST methods under the regulation and surveillance of Taiwan CDC. The agar proportion method on Middle- brook 7H10 (Creative Microbiologicals or Sancordon, Taipei, Taiwan) and BACTEC MGITTM 960 SIRE Kits with a liquid culture system were recommended. The critical drug concentrations for the agar proportion method on 7H10 are INH 0.2 μg/ml, INH 1.0 μg/ml, RMP 1.0 μg/ml, EMB 5.0 μg/ml and SM 2.0 μg/ml, while on MGIT 960 system, the recommended critical concentrations are INH 0.1 μg/ml, RMP 1.0 μg/ml, EMB 5.0 μg/ml and SM 1.0 μg/ml. Growth on the control medium is compared with the growth on the drug-containing medium to determine susceptibility. DST results were classified as resistant or susceptible. Tests were validated by the susceptibility of M. tuberculosis H37Rv included in the same DST.
6.Line 216-222 – It is very uncommon that Mtb strain is resistant to lower concentration of INH and sensitive to higher concentration. Do you have any explanation?
Reply: According to the DST of M. TB in our institute, resistance of low and high concentration of INH was determined. In our previous study (PLoS One 2014 Jan 22;9(1):e86316. Clinical characteristics and treatment outcomes of patients with low- and high-concentration isoniazid-monoresistant tuberculosis.), 134 of 1229 culture-positive tuberculosis patients were INH monoresistance. 44 patients were with INH low concentration (0.2 µg/mL) resistance and 90 patients were with high (1.0 µg/mL) concentration resistance. One-third (44/134) low INH concentration (0.2 µg/mL) resistance was reported.
7.Line 230 – The sentence does not have to start with the word "In addition"
Reply: We had deleted “In addition” on line 230 of the revised manuscript.
